# The Role of the Activator Additives Introduction Method in the Cold Sintering Process of ZnO Ceramics: CSP/SPS Approach

**DOI:** 10.3390/ma14216680

**Published:** 2021-11-05

**Authors:** Yurii D. Ivakin, Andrey V. Smirnov, Alexandra Yu. Kurmysheva, Andrey N. Kharlanov, Nestor Washington Solís Pinargote, Anton Smirnov, Sergey N. Grigoriev

**Affiliations:** 1Chemistry Department, M. V. Lomonosov Moscow State University, 119991 Moscow, Russia; ivakin@kge.msu.ru (Y.D.I.); Kharl@kge.msu.ru (A.N.K.); 2Mobile Solutions Engineering Center, MIREA-Russian Technological University, 119454 Moscow, Russia; smirnov_av@mirea.ru; 3Center for Design, Manufacturing and Materials, Skolkovo Institute of Science and Technology, Bolshoy Boulevard 30, build. 1, 121205 Moscow, Russia; 4Laboratory of Electric Current Assisted Sintering Technologies, Moscow State University of Technology “STANKIN”, Vadkovsky per. 1, 127055 Moscow, Russia; nw.solis@stankin.ru (N.W.S.P.); a.smirnov@stankin.ru (A.S.); s.grigoriev@stankin.ru (S.N.G.)

**Keywords:** oxide ceramics, zinc oxide, cold sintering process, thermovapor treatment, microstructure, spark plasma

## Abstract

The great prospects for introducing the cold sintering process (CSP) into industry determine the importance of finding approaches to reduce the processing time and mechanical pressure required to obtain dense ceramics using CSP. The introducing zinc acetate into the initial ZnO powder of methods, such as impregnation, thermovapor autoclave treatment (TVT), and direct injection of an aqueous solution into a die followed by cold sintering process using a spark plasma sintering unit, was studied. The effect of the introduction methods on the density and grain size of sintered ceramics was analyzed using SEM, dynamic light scattering, IR spectroscopy, and XRD. The impregnation method provides sintered samples with high relative density (over 0.90) and significant grain growth when sintered at 250 °C with a high heating rate of 100 °C/min, under a uniaxial pressure of 80 MPa in a vacuum, and a short isothermic dwell time (5 min). The TVT and aqueous solution direct injection methods showed lower relative densities (0.87 and 0.76, respectively) of CSP ZnO samples. Finally, the development of ideas about the processes occurring in an aqueous medium with CSP and TVT, which are subject to mechanical pressure, is presented.

## 1. Introduction

In recent decades, significant progress in the evolution of promising new sintering methods has been made. For example, certain methods, such as FLASH sintering [1], spark plasma sintering (SPS) technologies [2,3,4], modified SPS [5], and cold sintering processes [6] have been developed. Common to all these new approaches is the emphasis on reducing the sintering time and temperature, and this can be considered as an advantage that favors a reduction in energy consumption [7] and consequently the emission of CO_2_ into the atmosphere [8], which is one of the great problems for the planet. On the other hand, a significant reduction in temperature and sintering time allows the consolidation of volatile and metastable phases, and the obtaining of unconventional microstructures provides the possibility of producing new materials with improved functional properties. Thus, new sintering technologies can have a decisive impact on the fields of biomaterials, electronics, communication systems, nuclear waste encapsulation, energy production and storage, electromechanical devices, and catalysis [9].

One of the most important innovations of the last decade has been the cold sintering process (CSP) [6]. Baba Heidary et al. [10] demonstrated that a reduction of sintering temperature and energy consumption during the cold sintering process can reach hundreds of degrees Celsius and two orders of magnitude, respectively, compared to traditional high-temperature sintering processes.

The principle of CSP could be summarized as a powder mixture (generally ceramic oxide) that, in combination with a liquid phase (generally water), is uniaxially pressed (50–500 MPa) in a die and sintered at a temperature that is generally below 350 °C [11]. CSP differs from the well-known hydrothermal hot pressing (HHP) process [12] since the latter is limited to obtaining a comparatively narrow nomenclature of ceramic materials (silicon oxide-based materials, hydroxyapatite based bioceramics, zirconium oxide, and radioactive waste [9]), which are formed during hydrothermal chemical reactions at high mechanical pressure and with long residence times (up to several tens of hours).

CSP can be considered as a significant evolution of the HHP process, although both methods have common features: using an aqueous medium, uniaxial pressing, and temperatures below 500 °C. The key difference between CSP and HHP is that in CSP a sealed reactor is not used (the sample does not experience hydrothermal conditions). Thus, the liquid is a metastable transition phase, and is squeezed out and evaporates from the working volume over time. From this point of view, the difference between CSP and HHP is purely technological, and both processes probably have common stages in terms of the mechanism [6,9].

Unlike HPP, CSP is a universal process in terms of the breadth of materials obtained. Currently, the main possibility of sintering of more than 50 material compositions (mainly based on ceramic oxides) using the cold sintering process has been experimentally confirmed. These material compositions include microwave dielectrics, ferroelectrics, structural ceramics, lithium ion cathodes, solid state electrolytes, semiconductors, ceramic adhesives, magnetic ceramics, and inorganic glass [13].

The most important scientific problem is the lack of generally accepted and experimentally confirmed ideas about the mechanisms of CSP. From a practical point of view, the potential compatibility of CSP with large-scale industrial productions is of great importance. Despite a lack of understanding of the mechanisms of the CSP, attempts are being made to overcome the technological problems related with the use of long-term dwelling under high mechanical pressures in order to obtain dense ceramics parts [13]. The two tasks can be distinguished to solve this issue: reducing the applied mechanical pressure (or eliminating it) and reducing the CSP time. Gonzalez-Julian et al. [14] showed the fundamental possibility of using SPS machines for CSP in order to obtain dense ceramics at low temperatures in a short time, using high heating rates (100 °C/min) and a short dwell time (5 min). However, the CSP in these conditions required a pressure greater than 100 MPa, which requires the use of a special steel die, since the standard graphite die did not allow it to withstand mechanical pressures of that magnitude.

Bang et al. [15] focused on a size scale-up CSP demonstrated the fundamental possibility of obtaining ceramics and composites based on ZnO, Li_1.5_Al_0.5_Ge_1.5_(PO_4_)_3_ and zeolite using CSP at a pressure below 50 MPa, but with long dwell times from 15 to 480 min. This result is an important step in the industrial implementation of the CSP method. However, an increase in the size of the samples led to microstructural inhomogeneity, which can be caused by randomly localized evaporation of the transition liquid phase. Currently, the search for approaches to simultaneously accelerate CSP and reduce the mechanical pressure continues to be an urgent topic of research.

The most detailed data on the effect of the type and amount of the activating additive, temperature, pressure, and powder characteristics on the density and grain size of cold sintered ceramics are available for ZnO [14,15,16,17,18,19,20,21,22]. However, the degree of influence of the activator’s introduction method, when using water as a transport phase, remains poorly studied. In our previous work [20], a high mechanical pressure (up to 396 MPa at the compaction stage and 77 MPa at dwell stage), slow heating rate (250 °C during 40 min.), and long dwell time (1 h) were used. Furthermore, it was shown that the introduction of zinc acetate by both ZnO powder impregnation method and the traditional method of acetate introduction in an aqueous solution does not change the CSP results while maintaining identical amounts of activator and water.

In addition, in another previous work [21] it was shown that there is a similarity in the conditions of the crystal structure transformation of ZnO in both the water vapor medium at CSP and the autoclave thermovapor treatment (TVT).

The study of the oxide crystal formation process in pre- and supercritical aqueous media led to the idea of the appearance of solid-phase mobility of the crystal structure under TVT conditions in the presence of an activator. This additive facilitates the interaction of crystals with the medium. The solid-phase mobility appears due to exchange processes between crystals and the medium [21,23]. At the same time, mass transfer and crystal growth begin. The interaction of the activating additive with the initial powder during the TVT process leads to the surface and volume changes in the powder particles crystal structure [21,23,24]. The use of a raw material previously subjected to TVT can drastically change the results obtained after CSP compared to other traditional methods of activator introduction that are applied just before starting the CSP, such as direct powder mixing with water solution [22]; mixing ceramic and activator powders with subsequent water addition [25], and direct solution injecting into the die [14]. The study of the process features in such experimental conditions is important both in the context of the applied aspect and for the development of ideas about the existing CSP mechanisms. Guo et al. [6] proposed a CSP mechanism for ceramics densification that has received greater recognition in the scientific community and whose the main stage is mass transfer by dissolution-precipitation [10,12]. According to this model, three stages of compaction under the influence of mechanical pressure are considered: (i) the rearrangement of particles, which is facilitated by the liquid phase; (ii) grain growth; and (iii) pore removal. Stages (ii) and (iii) occur due to the dissolution of the solid phase in areas with high mechanical stress of contacting particles, followed by deposition on contactless surfaces with low mechanical stress [6]. However, subsequent studies have shown that more complex mechanisms are involved in the process of cold sintering, which is not yet fully understood [14,26]. Afterwards, it was noted that there is a problem in the re-deposition model: dissolution and precipitation are exclusively surface events and can only lead to particle enlargement but cannot contribute to their compaction [14,21]. Guo et al. [12] established that the liquid volume necessary for the formation of ceramics in CSP is only approximately 5 vol.% of powder, and the liquid was called “The transport phase”, since the analogy of liquid-phase sintering. Gonzalez-Julian et al. [14] revealed that ZnO sintering occurs in the presence of 1.6 wt.% water, which corresponds to only two water monolayers on the particles surface and it is not enough for the dissolution-precipitation processes. Moreover, the authors noted that half of 1.6 wt.% water dissociates and diffuses into the oxide lattice, increasing the defectiveness of crystals and grain boundaries. Floyd et al. [27] demonstrated that during CSP of ZnO in the presence of the crystalline additive Zn(Ac)_2_·2H_2_O, used as the transport phase, the mass transfer is not associated with the added liquid water, but is related to the adsorbed water and structural water of the additive. In addition to this, the authors did not consider the action mechanism of the transport phase but supposed that it enhances surface diffusion and grain growth. Considering the above this effect, the compaction during CSP was explained as a decrease in the diffusion activation energy of atoms along grain boundaries due to the high concentration of hydroxyl ions and other defects formed due to dissociative adsorption of water [14,26,27,28,29]. Further, starting with the work [30], the mass transfer due to surface diffusion is actively discussed in the literature. As a result, when describing the mass transfer stage in the CSP model, the emphasis was shifted from the dissolution-precipitation mechanism to the surface diffusion mechanism. In addition, in our works [21,23], the explanation of the CSP mechanism is based on the idea of the solid-phase mobility of the crystal structure, which occurs due to the intensive exchange of water and activator molecules between the environment and the dissociated forms of these molecules bound in the volume of crystals. To further develop the CSP mechanism’s understanding, it is important to obtain, systematize, and analyze as much new experimental data as possible. In particular, to study the effects of various CSP conditions.

The present work aims to study the effect of the activator’s introduction method on the grain size and density of ZnO ceramics obtained in a short time and under a lower-pressure CSP using SPS equipment. The CSP/SPS experiments were carried out under the same composition and density conditions of the aqueous medium studied in previous works [20,21,23], but with a significantly higher heating rate and a short dwell time. The durations of the heating and holding stages (40 and 60 min, respectively, were used in [20,21,23]) differed by an order of magnitude compared with the values used in this study (2 and 5 min, respectively). Comparing the results obtained under conditions of a significant change in the time factor reveals the role of different stages of the CSP mass transport mechanism. In addition, reducing the CSP time is important in reducing the process’s energy consumption. The obtained results also revealed the role of the activator additives introduction method on the CSP/SPS process features. This paper describes for the first time the results of using TVT as a method of introducing an activator additive into ZnO powder for CSP. In previous works [20,21,23] and this study, the same initial ZnO powder and other reagents were used. The impregnated powder and TVT samples were prepared each time anew immediately before the experiment.

## 2. Materials and Methods

### 2.1. Composites Preparation

In the present work ZnO powder (JSC “St. Petersburg Red Chemist”, Russia) with a mean particle size of 0.2 μm and solid zinc acetate Zn(CH_3_COO)_2_·2H_2_O (LLC “Prime Chemicals Group”, Russia) were used as starting materials. All reagents had a purity of the main component of >99 wt.% according to the supplier’s documentation.

The solid zinc acetate Zn(Ac)_2_ activating additive 0.5 wt.% or 0.185 mol%, 2.5 wt.% or 0.927 mol%, and 5 wt.% or 1.853 mol% was introduced into the raw powder in three ways:injecting the activator solution using a syringe directly into the die with pre-pressed ZnO powder, similar to [14];preliminary application by impregnation method;TVT of ZnO powder previously impregnated by Zn(Ac)_2_.

When Zn(Ac)_2_ was applied by impregnation, 20 g of ZnO powder was mixed during treatment in an ultrasonic bath with 30 mL of an activator solution in distilled water. The obtained wet mixture was dried at 70 °C for 12 h. The resulting powder was ground in an agate mortar and subsequently passed through a 300-μm sieve.

TVT was performed in 17 mL laboratory autoclaves (self-made) using a copper gasket and Teflon containers with a loose-fitting lid. The container was installed in an autoclave on a stand that the water does not get into the container with dry powder, and the process goes only in water vapor. Water was poured into the bottom of the autoclave below a container’s stand. 1–5 g of ZnO powder previously impregnated by Zn(Ac)_2_ was placed in the container. The autoclave was sealed, heated in a furnace to 220 °C, and held for 20 h. With such loading of the autoclave, heating and isothermal dwelling of ZnO occurred in a water vapor environment—under TVT conditions. The peculiarities of TVT are described in [21]. The content of the additive in the reaction medium was calculated in wt.% relative to ZnO.

### 2.2. Cold Sintering Process

CSP was carried out using the SPS technique H-HP D 25 (FCT Systeme, GmbH, Rauenstein, Germany). The powder pre-pressing was realized on the 12 tons Manual Hydraulic Press Test System (Carver, Inc., Wabash, IN, USA). Graphite punches and die with an inner diameter of 20 mm, and graphite foil were used. For reproducibility of control of the heating conditions in each experiment, a thermocouple was installed in a die wall a distance of 5 mm from the inner wall of die.

CSP samples preparation: the initial ZnO powder was mixed with deionized water similar to the procedure described in the article [15]. For this, 3 g of ZnO powder was poured into a die and prepressed on a hand press with a pressure of 16 MPa, the upper punch was removed, and then in the center of the pre-pressed powder pellet with a thickness of 0.34–0.50 mm the 0.05 mL (1.6 wt.%) deionized water was added using a micro syringe (injecting). If possible, the drop was directed to the center of the pre-pressed pellet. Next, a graphite foil disk and a punch were placed into the die. The assembled die was transferred to the SPS machine, and the CSP was started according to the program:Pressing up to 15 MPa.Evacuation process (0.23 mbar vacuum).Pressing up to 50 MPa.Heating rate 100 °C/min.Pressing up to 80 MPa.Dwell time for 5 min at 80 MPa and a temperature of 250 °C.Cooling at a pressure of 15 MPa.Venting, extracting the mold.

The limiting pressure value is determined by the mechanical restrictions of the standard graphite tools, for instance, in [15] a steel die and punches were used for pressing at 100, 125, and 150 MPa.

### 2.3. Density Measurements

The calculation of the relative density was carried out according to the formula adopted from [25]:(1)$$d_{rel}\left(t\right)=\frac{m_{f}}{\pi \cdot r_{f}^{2}\cdot (t_{f}+\left(s_{max}-s\left(t\right))\right)\cdot \rho_{theo}}$$
where *m_f_*, *r_f_*, and *t_f_*—are, respectively, the mass, the radius, and the thickness of the final pellet; *s_max_*, *s*(*t*)—the maximum shrinkage of sample and shrinkage at a certain point in time, respectively (according to process data from SPS setup); *ρ_theo_*—theoretical density of ZnO equal to 5.61 g/cm^3^. The diameter and thickness of the samples were measured with a caliper and a micrometer after cleaning from graphite foil (Figure 1).

### 2.4. XRD Characterization

XRD measurements of the powders and sintered samples (XRD-6000, Shimadzu Corp., Kyoto, Japan) were performed at diffraction angles of 2θ ranging from 5° to 60° (step scanning mode, step size 0.02°). Qualitative analysis was carried out by comparing experimental patterns with data from the database. The conclusion about the presence of layered basic zinc acetate (LBZA) was made according to [31].

### 2.5. Microstructural Characterization

Morphologic properties of the powder particles and sintered samples were evaluated by scanning electron microscope (SEM) JSM-6390 LA (JEOL Ltd., Tokyo, Japan) in the backscattered measurement mode. Particle size analysis was carried out by dynamic light scattering (DLS) in a Photocor Compact equipment (LLC Photocor, Moscow, Russia) after preliminary ultrasonic deagglomeration of an aqueous suspension within two minutes. The particle size distribution (PSD) was performed by SEM images analysis using the lognormal particle size distribution function [32]. Mean particle size D was determined by simply averaging over all measured objects in the sample. The Image-Pro software (Media Cybernetics, Rockville, MD, USA) was used to determine the size of 2000 to 3000 particles in each sample. The state change of the ZnO crystals after the Zn(Ac)_2_ addition was monitored by diffuse reflectance infrared Fourier transform spectroscopy (DRIFTS) in the range 400–7500 cm^−1^ (IR Fourier spectrometer EQUINOX 55/S, Bruker Co., Billerica, MA, USA).

## 3. Results

The results of experimental studies are presented in the following sequence: first, the characterization of the raw ZnO powder is presented, then it is followed by the effect description of two activator introduction methods (impregnation and impregnation + subsequent TVT) and activator concentrations on the raw powder size, morphology, and phase composition; second the characteristics of ZnO ceramic samples obtained by CSP from the powders described in the previous subsection are presented.

### 3.1. The Effect of Activator Introduction Method on the Initial ZnO Powder Characteristics

Summary data on measuring the particle sizes of ZnO powders after their impregnation with Zn(Ac)_2_ and after impregnation with subsequent TVT are presented in Table 1.

The difference in the average powder particle sizes, determined by two different methods, may cause the hydrodynamic diameter of the particles to be determined by DLS method and the influence of the residual agglomerates remaining after ultrasonic deagglomeration. In addition, the average size characteristic of DLS, d_50_ does not take into account changes in the levels of small and large particle sizes. Initial ZnO powder particles morphology is shown in Figure 2a. The average crystal size determined by SEM image analysis and DLS were D = 0.193 μm and d_50_ = 0.210 μm, respectively. PSD is described by a lognormal function and consists of a single component. After applying 0.5 wt.% Zn(Ac)_2_ (0.5%Ac/ZnO powder sample), the size range did not change, but there were a differentiation of the crystals with the appearance of two components of the PSD (Figure 2c,d) and a slight decrease in mean particle size D (Table 1) due to the appearance of a fine component PSD. The impregnation of the ZnO powder by 0.5 wt.% Zn(Ac)_2_ and the subsequent TVT also lead to a change in the PSD. However, during the TVT process, a complete separation of PSD into a finely dispersed component and the main group of crystals (powder particles) occurs, the mass transfer between which leads to the appearance of new components and an increase in the mean particle size D (Figure 2f and Figure 3d). Moreover, the transfer of mass to the crystals of the finely dispersed component does not occur. With an increase in the content of Zn(Ac) up to 5 wt.%, the separation of the fine component becomes more pronounced after impregnation, and the main proportion of crystals is divided into components without changing the mean particle size D (Figure 3a,b). After TVT, the finely dispersed component is completely separated from the main part of the crystals due to the redistribution of mass between them with an increase in the mean particle size D and the number of PSD components (Figure 3c,d). The bimodal nature of PSD can have a significant influence on the relative density of CSP samples obtained from powder samples after TVT (TVT_0.5%Ac/ZnO, TVT_2.5%Ac/ZnO, TVT_5%Ac/ZnO) due to a denser packing of ZnO powder particles in a mold [33,34]. For samples impregnated without subsequent TVT (0.5%Ac/ZnO, 2.5%Ac/ZnO, 5%Ac/ZnO), the nature of PSD will have a minor effect on the relative density of CSP samples. Thus, the differentiation of crystals into components during impregnation occurs due to the formation of crystals of a finely dispersed component, and mass transfer is involved in the differentiation of the rest of the crystals during TVT. The change in the mean particle size D during impregnation and TVT is shown in Figure 4. It can be seen that as a result of the redistribution of mass at TVT, that the mean particle size D increases more than twice regarding the initial ZnO powder.

Figure 5 shows XRD diagrams of ZnO samples with 0.5, 2.5, and 5 wt.% Zn(Ac)_2_. After applying Zn(Ac)_2_, Zn_5_(OH)_8_(CH_3_COO)_2_·2H_2_O (layered basic zinc acetate (LBZA) [31] reflexes appear on the XRD diagrams of ZnO powder samples. The intensity of reflexes in the range of 2θ 5–30°, related to LBZA, increased by 10 times. The intensity of the main reflex of 2θ 6.5° is proportional to the amount of acetate applied, while the reflex at 2θ~13° practically does not depend on its concentration. This may be caused to the lamellar morphology of LBZA and the variable size of the plates with a small change in their thickness. The disappearance of the LBZA reflexes indicates its decomposition during TVT and CSP. The ZnO reflexes in the range of 2θ 30–60° are the same for all samples shown in Figure 5a–c.

Figure 6 shows the DRIFTS spectra of ZnO powder in the initial state, after application by impregnation of 5% Zn(Ac)_2_ (5%Ac/ZnO powder sample) and after impregnation with subsequent TVT (TVT_5%Ac/ZnO powder sample). The initial ZnO powder absorption bands in the region of 1000–7500 cm^−1^ refer to the adsorbed components of the atmosphere in which the powder was stored. The application of Zn(Ac)_2_ by the impregnation method leads to the appearance in the samples DRIFTS spectra of several absorption bands, which were absent from the initial ZnO powder (Figure 6). Absorption bands in the region of 4000–7500 cm^−1^ are not described in the literature. Valence vibrations of hydroxyl groups cause the absorption in a wide area with a maximum of about 3363 cm^−1^. After introducing of 5% Zn(Ac)_2_ by both methods, the absorption in this area increases noticeably. The increase in absorption is associated with the hydroxylation of ZnO due to the dissociative adsorption of water molecules [35]. The bands of valence vibration of CH bonds at 2933 and 3011 cm^−1^ appear after impregnation of 5% Zn(Ac)_2_ and impregnation with subsequent TVT. They are displaced relative to the usual positions of 2850 and 2920 cm^−1^ in Zn(Ac)_2_ [36]. The displacement of the peak positions is associated with the influence of the solid matrix on the state of the bonds and indicates the localization of these groupings inside the crystals. The peaks of the complex absorption band (1336–1560 cm^−1^) in the range of 1300–1600 cm^−1^ belong to the carboxylate group [37]. It should be noted that the width of these bands decreases after TVT, while the absorption bands at 5125 and 6980 cm^−1^, on the contrary, become wider.

The main result of TVT is the destruction of LBZA, the growth of crystals, and the improvement of the crystal ZnO structure, at which the content of hydroxyl groups decreases [38]. The decomposition of LBZA, observed according to the XRD analysis (Figure 5b), leads to a decrease in the intensity of the corresponding absorption bands, but all the bands are preserved in the IR spectrum. Consequently, traces of acetate remain in the crystal structure.

Figure 7 shows the spectra of additional absorption caused by a change in the Zn(Ac)_2_ content in samples that have undergone TVT. The additional absorption spectra are obtained by subtracting the spectrum of a sample with a lower Zn(Ac)_2_ content from the spectrum of a sample containing more Zn(Ac)_2_: ΔR = R(TVT_5%Ac/ZnO)-R(TVT_2.5%Ac/ZnO) и ΔR = R(TVT_2.5%Ac/ZnO)-R(TVT_0.5%Ac/ZnO).

It can be seen that with an increase in the Zn(Ac)_2_ content, the position of a number of bands remains mainly in the region of more than 4000 cm^−1^. The position of the bands in the regions belonging to the hydroxyl (3000–4000 cm^−1^) and carboxylate (1200–1600 cm^−1^) groups changes. This means that in the first case, with an increase in the Zn(Ac)_2_ content, the spectrum of states of the absorbing groups does not change. In the second case, the shift of the absorption bands reflects a change in the localization conditions of additional groupings in the ZnO crystal structure.

### 3.2. The Effect of Activator Introduction Method on the ZnO CSP/SPS Process

Table 2 presents data on the average grain size and relative density of CSP samples. The mean powder particle size D (data from Table 1) and mean grain size G of ceramic samples are given according to the results of the analysis of the SEM images.

SEM images of samples microstructures obtained from impregnated powders and subjected to TVT are shown in Figure 8.

The relative density of samples increases with the increasing Zn(Ac)_2_ concentration both when using the impregnation method and TVT. At the same time, the average grain size has different tendencies to change depending on the Zn(Ac)_2_ concentration for different activator introduction methods. When using TVT, there is a direct dependence, and when impregnating, the opposite is observed. The Zn(Ac)_2_ introduction in an aqueous solution with a syringe directly into a mold with a pre-pressed powder (sample CS_ZnO + H_2_O-0.5%Ac) does not give a pronounced effect with respect to the grain size, since there is actually a marked effect in terms of relative density under the conditions of this experiment (pressure 80 MPa, heating–dwell cycle about 7 min), even in comparison with the sample (CS_ZnO + H_2_O) obtained using deionized water without any Zn(Ac)_2_ addition (Figure 9). However, although the range of grain sizes and their average size G do not change, there are differences in the distribution of grain sizes into components. Samples with different introduction methods and the same amount of activating additive—CS_0.5%Ac/ZnO and CS_ZnO + H_2_O-0.5%Ac differ especially strongly in microstructure and grain size (Figure 9). Probably, since after impregnation, Zn(Ac)_2_ is bound in the crystal structure, and after evaporation and extrusion of excess water from the mold, CSP continues in water vapor, almost without reducing activator content. When the solution is added to the mold after its partial evaporation and extrusion in a place with water, a certain amount of Zn(Ac)_2_ is lost, and CSP continues under conditions of a reduced activator content, which leads to a restriction of grain growth and a decrease in relative density.

Figure 10 shows the time-dependent relative density curves for each CSP sample and the CSP/SPS process stages. The impregnated and impregnated + TVT groups of CSP samples are shown in different colors. The dependencies of the relative density on the sintering time have a non-monotonic character, reflecting the processes occurring with an increase in the temperature and pressure. The stepwise stages of compaction visible in Figure 10 require study with a more detailed registration. At the moment, there are separate attempts to explain them in [25,27].

In comparison with the traditional approach to CSP, the SPS method has several distinctive features. An important stage of the process is the air evacuation process (vacuum). A stable process was achieved when vacuuming to 0.23 mbar pressure. At a higher pressure and, accordingly, a short evacuation time, the air is quickly removed from the mold, but the water does not have time to evaporate and occupies all the free space. Then, with rapid heating, the water expands almost instantly, leading to the mold’s destruction. At a pressure in the SPS chamber of 0.23 mbar, reached in 8 min, part of the water evaporates through the gaps between the graphite foil and the punches and, due to the freed space in the die, the vapor pressure is maintained, which is in equilibrium with liquid water. As a result, CSP flows stably partly in the medium of saturated water vapor and partly in the medium of preserved condensed water. The cooling stage is also crucial due to the rapid free cooling of the mold under SPS conditions. In some experiments, cracks were formed during the CSP/SPS samples from impregnated ZnO powder (CS_2.5%Ac/ZnO and CS_5%Ac/ZnO). Additional studies are required to determine the optimal modes of the cooling stage.

## 4. Discussion

According to the data of Table 2 and Figure 8, the use of ZnO powder with pre-applied Zn(Ac)_2_ by the impregnation method leads to the best CSP results (under the conditions of this study). This fact can be explained by the appearance of solid-phase mobility of the impregnated ZnO powder’s crystal structure during a long time (12 h) drying at a 70 °C, as evidenced by changes in the PSD components. At the same time, according to IR spectroscopy data (Figure 6 and Figure 7), the interaction with water leads to the hydroxylation of ZnO crystals, the adsorption of acetate ions, ions destruction into smaller groups, and subsequent diffusion into the volume of ZnO crystals. Thus, solid-phase mobility appears due to the exchange of water molecules and acetate ions between crystals and the environment [21]. At the impregnation temperature of ~70 °C, the exchange rate is low, the solid-phase mobility of the structure and the processes associated with it are in the initial stage of development. With TVT, the processes of exchange with the environment and the solid-phase mobility and transformation of the crystal structure are significantly accelerated. A similar acceleration occurs under CSP conditions but with the action of mechanical pressure.

The time-dependent relative density curves of the ZnO CSP samples have a complex form (Figure 10). The non-monotonic character is associated with the consumption of thermal energy for the processes taking place, the main of which are water evaporation, decomposition of the additive Zn(Ac)_2_, and sintering, which includes mass transfer, and, probably, the solid-phase mobility of the ZnO crystal structure. In all cases, due to the ratio of a small amount of water (1.6%) and the mass of ZnO powder, the effect of water evaporation is small and can be ignored. In the samples CS_ZnO + H_2_O and ZnO + 0.5% Ac-H_2_O, densification is minimal, mechanical pressure leads to compaction of the powders, then with an increase in temperature and during dwelling, the density does not change or increases slightly (ZnO + 0.5%Ac-H_2_O). In the group of samples after TVT, only traces of Zn(Ac)_2_ are present, and the effect of its decomposition on the dependencies of density changes is not observed. The kinks of dependencies reflect the sintering processes. However, they are affected by the slow setting of the temperature. This is due to the fact that the mold heats up quickly, and the powder mass (3 g) warms up more slowly, and as it heats up, the stages of increasing the density of samples are activated.

Due to the preliminary activation of the crystal structure under impregnation conditions, the acceleration of mass transfer in a short time of CSP/SPS (used in this work) leads to the formation of a dense microstructure of ceramic samples (Figure 8d–f). The decrease in the size of ceramic grains (samples CS_0.5%Ac/ZnO, CS_2.5%Ac/ZnO, CS_5%Ac/ZnO in Table 2) is probably due to the inhibition of mass transfer by the impurity phase of the LBZA, the lifetime of which (between formation and decomposition) increases with an increase in the Zn(Ac)_2_ content in the initial ZnO powder. However, under these conditions, the solid-phase mobility of the crystal structure and the ability to change their shape (deformation) increases. As a result, a dense microstructure with small grains is formed.

During a long period of TVT (20 h in this work), the crystal structure of ZnO powder is transformed under the action of solid-phase mobility: the crystal size of powder samples increases. In addition, solid-phase mobility leads to a decrease in the number of defects in the crystal structure. As a result of the improvement of the crystal structure, its solid-phase mobility decreases, and crystal growth stops. The conclusion about improving the structure at a certain stage of TVT is made based on several signs. When crystals are formed under these conditions, the sizes of coherent scattering regions increase [39,40,41] the content of hydroxyl groups decreases [41,42,43,44] and well-defined crystals with flat sides and sharp edges appear [41,42]. CSP at 250 °C with the addition of deionized water leads to compaction without noticeable signs of powder particles sintering after TVT (samples CS_TVT-0.5%Ac/ZnO and CS_TVT-2.5%Ac/ZnO (Figure 8a,b, Table 2)). However, when the Zn(Ac)_2_ content increases from 0.5 up to 5% in the crystal structure of ZnO powder after TVT, more residual activator forms (LBZA) remain (Figure 5a), which leads to their release into the aqueous medium and the appearance of exchange processes between the medium and ZnO crystals in the CSP process. Exchange processes with the medium activate solid-state structural mobility and mass transfer, which leads to the increased relative density of ceramic samples (Figure 8a–c, Table 2). Furthermore, under these conditions, the LBZA phase is absent, and the growth of ceramic grains is observed (Table 2). As a result, the sample CS_TVT-5%Ac/ZnO (Figure 8c) is characterized by significant grain growth and dense microstructure formed in a short time CSP, which shows a high intensity of exchange of molecular groups between crystals and the aqueous medium.

CSP ZnO without an activator with the addition of deionized water (sample CS_ZnO + H_2_O) does not lead to rapid densification (Figure 9a). A change in the PSD compared to Figure 2a, shows that the differentiation of crystals into components begins in these conditions, but the size range and average size D of the crystals do not change. Long dwell time under these conditions may lead to densification caused by activating influence of some technological impurities (acids and oxidizing agents) in graphite foil, which could get into the foil during manufacture process [18].

The weak sintering effect of sample CS_ZnO + H_2_O-0.5%Ac with the injection of 0.5% Zn(Ac)_2_ solution directly to the mold (Figure 9b) can be associated with the short time of the CSP, during which the solid-state structural mobility did not do so have time to develop. However, the differentiation of crystals took place to a greater extent than in a water medium without an activator (Figure 9a).

A rapid process in the CSP/SPS reveals a difference in the influence of the solid-state structure mobility of the mass transfer. In long-term CSP processes (without SPS rapid heating), no difference between impregnation or injecting methods of additive introduction was observed [20]. The effect of changing the method of Zn(Ac)_2_ introducing and its concentration under CSP/SPS conditions, taking into account the mechanical pressure and the holding time, can be estimated compared to the results obtained by the other authors (Table 3). It is worth noting that in [14] a mechanical pressure of 150 MPa was required to achieve a relative density of 0.97 compared to 80 MPa used in this study to achieve the same density. This result is due to the fact that an increase in mechanical pressure accelerates surface diffusion due to the formation of defects caused by gradients of mechanical stresses [14]. Furthermore, the high density achieved in work [18] for CSP in a water medium for a long time without an additive can be associated with the action of technological impurities released from graphite foil, which is mentioned above.

The results of comparing the data in Table 3 lead to three practically important statements and assumptions:The impregnation of ZnO powder by Zn(Ac)_2_ allows reducing the mechanical pressure from 150 MPa to at least 80 MPa while maintaining the same CSP/SPS process short time (about 7 min) and achieving the high relative density of 0.97 as in work [14], without using a special expensive high-strength mold. As a result, it becomes possible to effectively use standard SPS tools for CSP research under controlled conditions and with the possibility to vary the heating speed, holding temperature, and cooling conditions within a wide range, which is difficult or completely inaccessible when using specially assembled laboratory setups for CSP based on hand presses.An increase in the content of Zn(Ac)_2_ introduced by the impregnation method in the range of 0.5, 2.5, 5 wt.% leads to an increase in the relative density of CSP samples and a simultaneous decrease in grain growth (Table 2 and Table 3). This result shows the possibility to control the microstructure of ceramics due to the selection method of additive introduction even in a short time CSP/SPS approach.The experimentally established possibility to reduce the pressure and the CSP time while maintaining a high relative density of ceramics due to the selection of additives introduction method have a substantial practical impact for further applied research aimed at industrial applicability of CSP.

## 5. Conclusions

The influence of the Zn(Ac)_2_ introducing method on the relative density and grain size of ZnO ceramic samples obtained by CSP using rapid heating at an SPS equipment was studied. It was shown that the method of impregnation of the initial powder with 0.5–5 wt% Zn(Ac)_2_ allows implementing CSP at reduced mechanical pressure. ZnO was compacted to a high relative density (over 0.90) with a pronounced grain growth under CSP/SPS conditions at a reduced pressure of 80 MPa, a high heating rate of 100 °C/min, and a short dwell time of 5 min at a temperature of 250 °C. Due to the preliminary activation (saturation with defects) of the structure of ZnO crystalline powder particles under impregnation conditions, an acceleration of mass transfer in a short time of CSP/SPS is observed, which leads to the formation of a dense microstructure of ceramic samples. The decrease in the size of ceramic grains with an increase in the content of Zn(Ac)_2_ is probably due to the restriction of mass transfer by the impurity phase of the LBZA, the duration of which increases with the increase in the content of Zn(Ac)_2_ in the reaction medium. However, under these conditions, the ZnO crystal structure solid-phase mobility and the ability to change their shape increases. As a result, a dense microstructure with small grains was formed.

The introduction of Zn(Ac)_2_ using impregnation and subsequent TVT leads to the growth of ZnO powder particles, the ordering and improving of their structure, and a decrease in its solid-phase mobility. However, CSP of TVT ZnO powder under conditions of reduced pressure and rapid heating and short time dwell with the addition of deionized water leads to compaction without noticeable signs of particle sintering.

The weak effect of sintering ZnO with the addition of an aqueous solution of 0.5 wt.% Zn(Ac)_2_ directly into the mold caused to the short CSP time (about 7 min), during which the structural mobility did not have time to develop, and the low mechanical pressure, which is not enough to accelerate surface diffusion due to the formation of defects caused by mechanical stress gradients.

The ability to reduce the mechanical pressure and CSP time while maintaining a high relative density of ceramics using the selection of additives introduction method has a substantial practical impact for further applied research aimed at industrial applicability of CSP.

## Figures and Tables

**Figure 1 materials-14-06680-f001:**
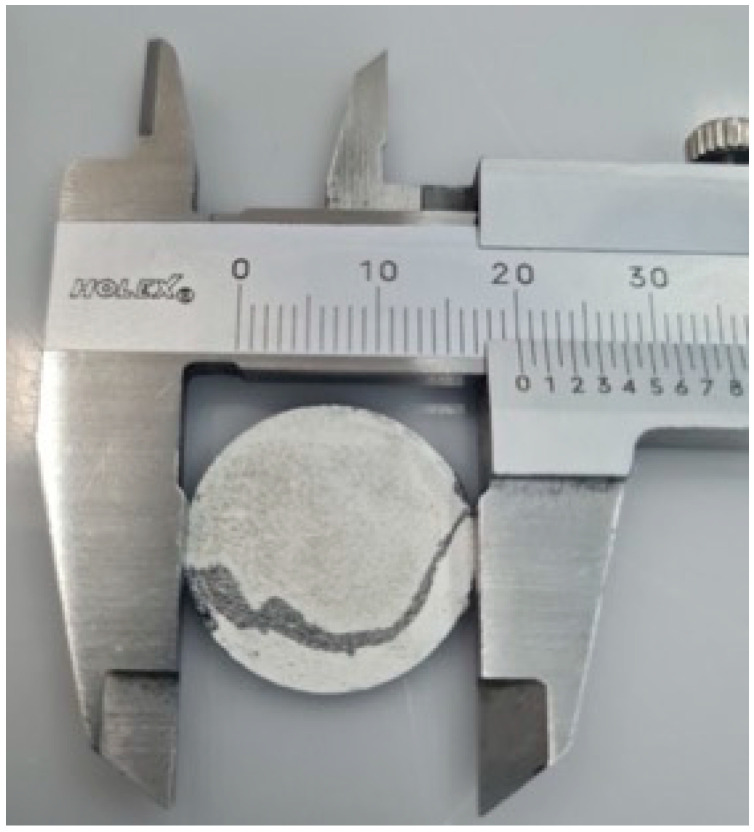
ZnO CSP sample.

**Figure 2 materials-14-06680-f002:**
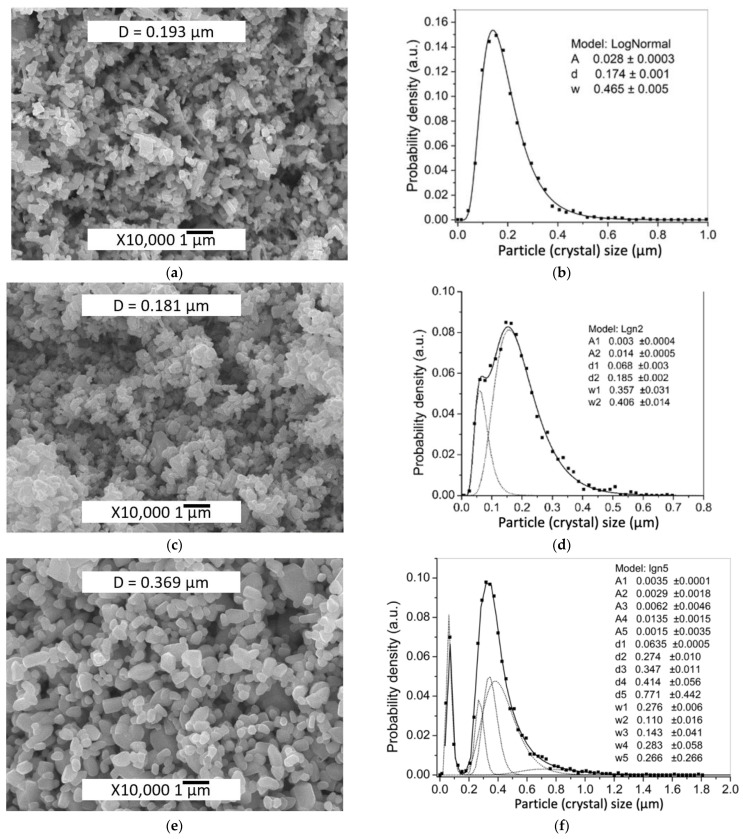
SEM image and PSD of ZnO powder in the initial state (**a**,**b**) after applying 0.5%Zn(Ac)_2_ by impregnation (0.5%Ac/ZnO powder sample) (**c**,**d**), and applying 0.5%Zn(Ac)_2_ by impregnation with subsequent TVT (TVT_0.5%Ac/ZnO powder sample) (**e**,**f**). The PSD is calculated from the respective SEM images.

**Figure 3 materials-14-06680-f003:**
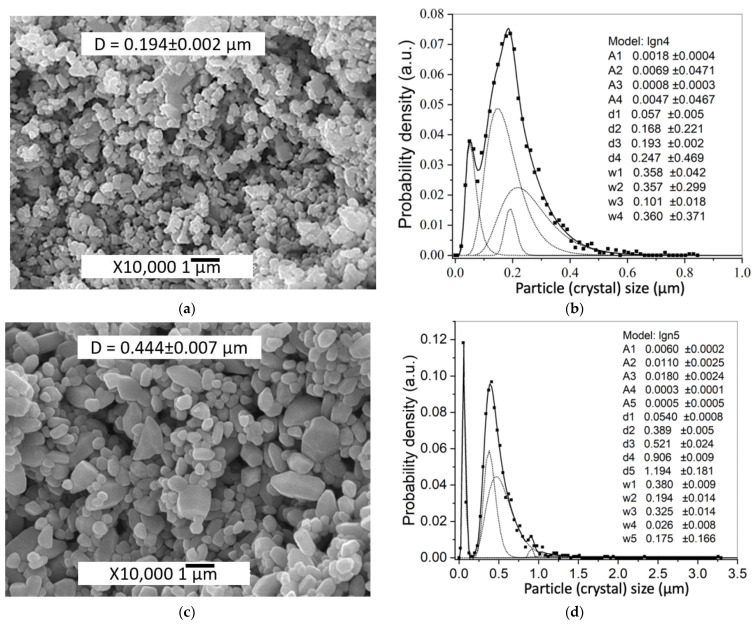
SEM images and PSD of 5%Ac/ZnO powder (**a**,**b**); and TVT_5%Ac/ZnO powder (**c**,**d**). The PSD is calculated from the respective SEM images.

**Figure 4 materials-14-06680-f004:**
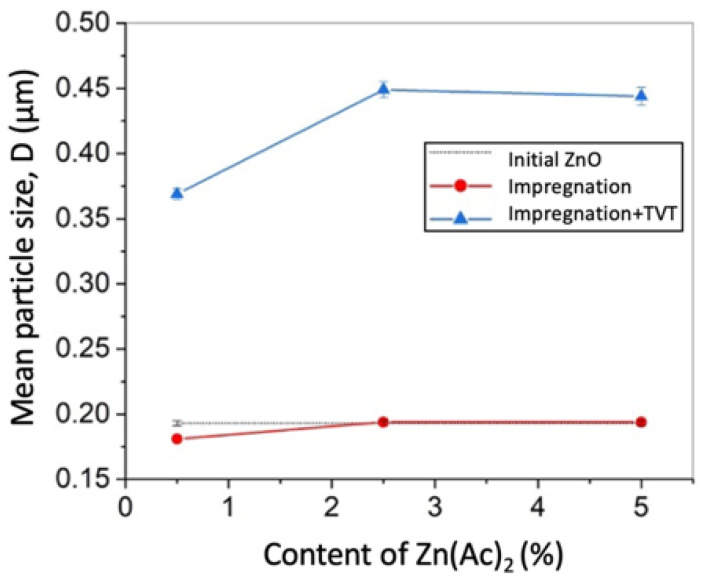
The dependence of the average crystal size on the content of Zn(Ac)_2_ during impregnation (1) and TVT (2): the dotted line—DLS method, the solid line—SEM images analysis method.

**Figure 5 materials-14-06680-f005:**
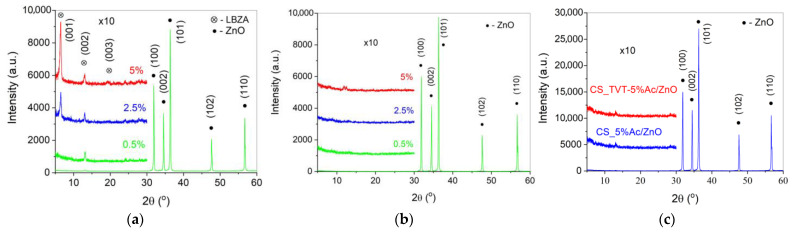
XRD diagrams of the powders after Zn(Ac)_2_ impregnation (**a**), after Zn(Ac)_2_ impregnation + TVT (**b**), and CSP samples (**c**). The reflexes intensity in the range of 2θ 5–30° increased by 10 times.

**Figure 6 materials-14-06680-f006:**
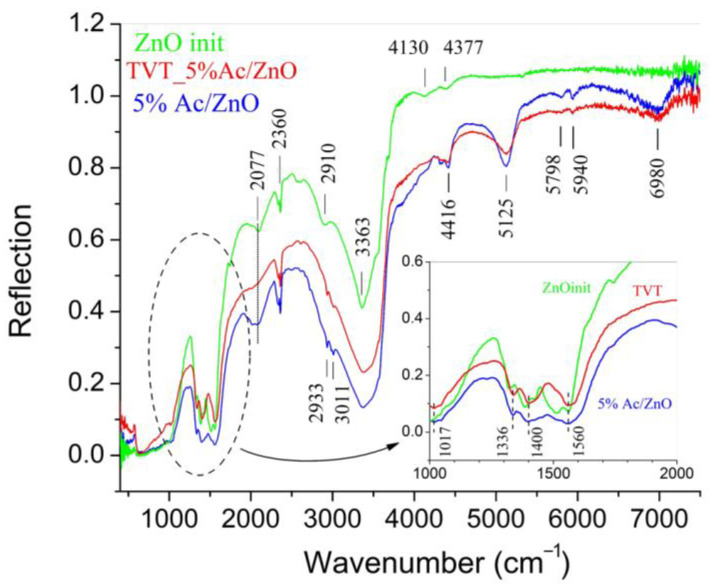
DRIFTS spectrum of ZnO powder in the initial state, after applying 5% Zn(Ac)_2_ by impregnation (5%Ac/ZnO) and impregnation + subsequent TVT (TVT_5%Ac/ZnO).

**Figure 7 materials-14-06680-f007:**
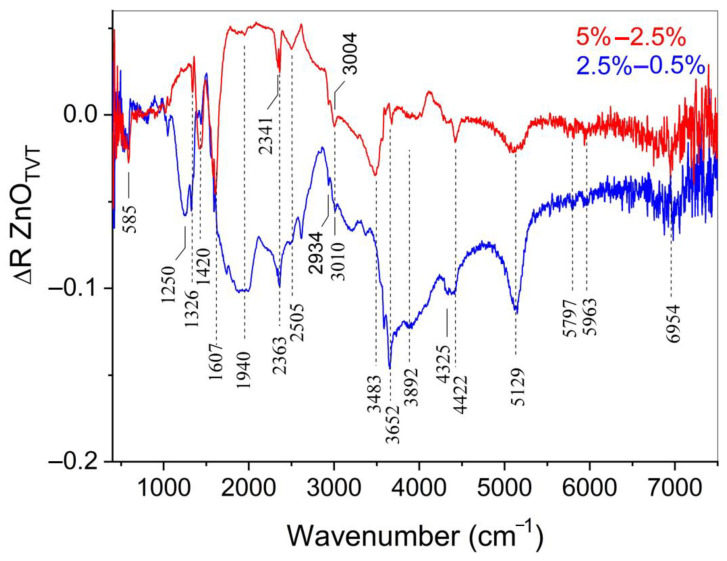
Additional absorption DRIFTS spectra in TVT ZnO powder samples. ΔR = TVT_5%Ac/ZnO-TVT_2.5%Ac/ZnO and TVT_2.5%Ac/ZnO-TVT_0.5%Ac/ZnO.

**Figure 8 materials-14-06680-f008:**
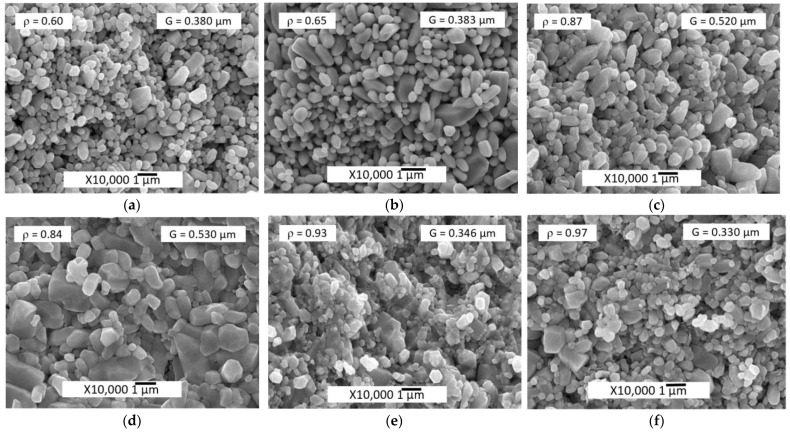
SEM images of the ZnO CSP samples microstructure: CS_TVT-0.5%Ac/ZnO (**a**), CS_TVT-2.5%Ac/ZnO (**b**), CS_TVT-5%Ac/ZnO (**c**), CS_0.5%Ac/ZnO (**d**), CS_2.5%Ac/ZnO (**e**), and CS_5%Ac/ZnO (**f**).

**Figure 9 materials-14-06680-f009:**
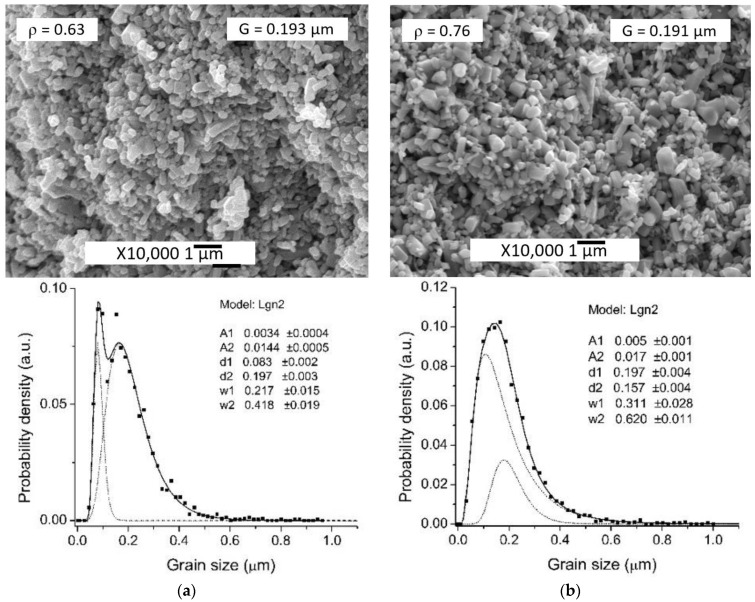
SEM images of microstructure and grain size distribution of the CS_ZnO + H_2_O (**a**) and CS_ZnO + H_2_O-0.5%Ac (**b**) CSP samples. The grain size distribution is calculated from the respective SEM images.

**Figure 10 materials-14-06680-f010:**
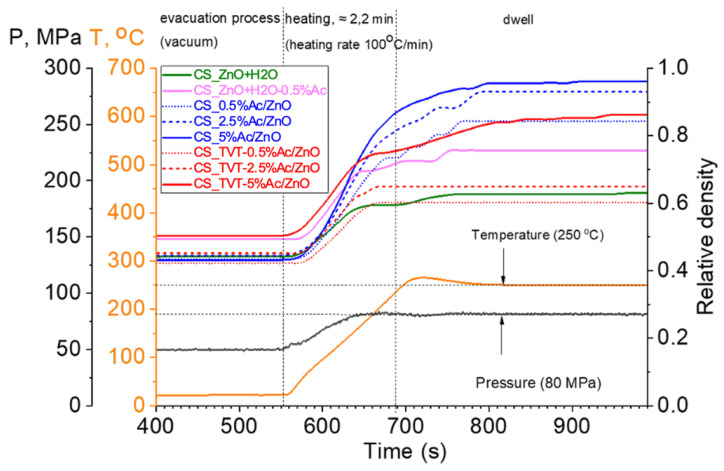
The time-dependent relative density curves of the ZnO CSP samples and the CSP/SPS process stages.

**Table 1 materials-14-06680-t001:** Powder particle sizes measured by DLS and SEM image analysis.

Sample	Zn(Ac)_2_ Content, wt%	Zn(Ac)_2_ Introduction Method	Mean Particle Size, μm	
			DLS, d_50_	SEM Image Analysis, D
ZnO initial	0	-	0.210 ± 0.004	0.193 ± 0.002
0.5%Ac/ZnO	0.5	Impregnation	0.228 ± 0.004	0.181 ± 0.002
2.5%Ac/ZnO	2.5		0.234 ± 0.005	0.194 ± 0.003
5%Ac/ZnO	5		0.245 ± 0.005	0.194 ± 0.002
TVT_0.5%Ac/ZnO	0.5	Impregnation + TVT	0.268 ± 0.005	0.369 ± 0.004
TVT_2.5%Ac/ZnO	2.5		0.291 ± 0.006	0.449 ± 0.006
TVT_5%Ac/ZnO	5		0.310 ± 0.006	0.449 ± 0.006

**Table 2 materials-14-06680-t002:** Relative density and grain sizes of CSP samples.

Sample	Zn(Ac)_2_ Introduction Method	Mean Powder Particle Size D, μm	Mean Grain Size G, μm	Relative Density *ρ*
CS_ZnO + H_2_O	injecting the DI water	0.193 ± 0.002	0.193 ± 0.002	0.63
CS_0.5%Ac/ZnO	impregnation	0.181 ± 0.002	0.530 ± 0.014	0.84
CS_2.5%Ac/ZnO		0.194 ± 0.003	0.346 ± 0.004	0.93
CS_5%Ac/ZnO		0.194 ± 0.002	0.330 ± 0.004	0.97
CS_TVT-0.5%Ac/ZnO	impregnation + TVT	0.369 ± 0.004	0.380 ± 0.004	0.60
CS_TVT-2.5%Ac/ZnO		0.449 ± 0.006	0.383 ± 0.01	0.65
CS_TVT-5%Ac/ZnO		0.444 ± 0.007	0.520 ± 0.007	0.87
CS_ZnO + H_2_O-0.5%Ac	injecting the solution	0.193 ± 0.002	0.191 ± 0.002	0.76

**Table 3 materials-14-06680-t003:** Effect of the additive introduction method on the characteristics of ZnO samples cold sintered at 250 °C using SPS equipment.

Heating Rate, °C/min	Dwell Time, min	Pressure, MPa	Powder Particle Size	Zn(Ac)_2_ Content, wt.%	DI Water Content, wt.%	Zn(Ac)_2_ Introduction Method	Mean Grain Size, μm	Relative Density	References
100	5	80	0.212 μm	0.5	1.6	impregnation (CS_0.5%Ac/ZnO)	0.530	0.84	This work
100	5	80	0.243 μm	5	1.6	impregnation (CS_5%Ac/ZnO)	0.330	0.97	This work
100	5	80	0.193 μm	0.5	1.6	injecting the solution (CS_ZnO + H_2_O-0.5%Ac)	0.191	0.76	This work
100	5	150	20–50 nm	0.5	1.6	injecting the solution	>0.100	0.97	[14]
20	60	300	40–100 nm	0	3.2	-	0.138	0.99	[18]

## Data Availability

The data presented in this study are available on request from the corresponding author after obtaining the permission of an authorized person.

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
