# Peer review of "The Role of the Activator Additives Introduction Method in the Cold Sintering Process of ZnO Ceramics: CSP/SPS Approach"

_materials, 2021, doi:10.3390/ma14216680_

Round 1
Reviewer 1 Report
The article Materials-1414401 "The role of the activator additives introduction method in the Cold Sintering Process of ZnO ceramics: CSP/SPS approach” by Yurii D. Ivakin et al. represents high scientific level. Presented manuscript is a follow-up of authors previous studies [20, 21, 27] concerning the Cold Sintering Process (CSP) of ZnO ceramics, and the novelty in this work is the comprehensive study of activator's introduction method influence on the grain size and density of ZnO ceramics obtained in a short time and less pressure CSP with the use of SPS equipment. In my opinion it is an interesting work that focuses on new ceramic sintering trends (CSP process), and has high research potential. Although, I found some issues that do not let me accept it for publication in the current shape. All of the aspects requiring explanations or corrections are listed below.
The individual manuscript sections are written more or less precisely, i.e.; introduction, discussion and conclusions are flawless, while the results section is in parts very chaotic and unclear. In addition, tables and figures should be reorganised to better fit the presented text. Authors should review the results section, and are asked to write in colour any improvement made in their revised text.
The main goal of article is to describe the different activator's introduction method and Cold Sintering Process, not the Spark Plasma Sintering. It is necessary to present SPS conditions, but Figure 1 is redundant.
According to the Authors, is the proposed method of density measurements applicable in the case of sintering with SPS device? Isn’t there some partially carbon diffusion from graphite foils that requires mechanical grinding to remove, thus limiting the sample diameter?
Table 3 shows some error. Position no.1 and no.3 refer to same material and method, although shows different mean grain size and density.
In addition, there are some smaller flaws like;
-there is no information on materials supplier,
-missing units or brackets for units in Figure 3b,
-Figure 7 has only indexes 7a and 7b, while in text (line 294) there is reference to Fig 7d.

Author Response
Dear Reviewer! Thank you for reviewing our manuscript. Here is the list of our response to your comments.
Reviewer 1
We thank the reviewer for his/her constructive comments to improve the presentation of materials in our paper. We have considered them to make our paper worthy of publication in the Materials
Reviewer 1 comments:
The individual manuscript sections are written more or less precisely, i.e.; introduction, discussion and conclusions are flawless, while the results section is in parts very chaotic and unclear. In addition, tables and figures should be reorganised to better fit the presented text. Authors should review the results section, and are asked to write in colour any improvement made in their revised text.
Authors´ answer
The results section was edited, changes in the text are marked.
Reviewer 1 comment:
The main goal of article is to describe the different activator's introduction method and Cold Sintering Process, not the Spark Plasma Sintering. It is necessary to present SPS conditions, but Figure 1 is redundant.
Authors´ answer
We agree with the view of the Reviewer, Figure 1 deleted.
Reviewer 1 comments:
According to the Authors, is the proposed method of density measurements applicable in the case of sintering with SPS device? Isn’t there some partially carbon diffusion from graphite foils that requires mechanical grinding to remove, thus limiting the sample diameter?
Authors´ answer
This method is applicable and easy to use. The method also makes it possible to determine the density of samples for which measurements by the Archimedean method are impossible (due to partial destruction and loss of their mass in the measuring environment). Diffusion of carbon from the graphite foil into the sample was not observed. Graphite foil could locally bake to the ceramic sample, and mechanical action was required to separate it. The diameter of the sample in this case did not change, it is due to the size of the used mold.
Reviewer 1 comments:
Table 3 shows some error. Position no.1 and no.3 refer to same material and method, although shows different mean grain size and density.
Authors´ answer
The error has been corrected.
Reviewer 1 comments:
In addition, there are some smaller flaws like;
-there is no information on materials supplier,
-missing units or brackets for units in Figure 3b,
-Figure 7 has only indexes 7a and 7b, while in text (line 294) there is reference to Fig 7d.
Authors´ answer
We have addressed all the comments and corrected the manuscript according your recommendations.
Reviewer 2 Report
Ms. Ref. No.: Materials-1414401
Title: The role of the activator additives introduction method in the Cold Sintering Process of ZnO ceramics: CSP/SPS approach
Yurii D. Ivakin et al.
Overall evaluation of the work;
I have reviewed this the manuscript by Ivakin et al.
In this study, the authors have investigated the effect of different mixing methods for ZnO and Zinc Acetate (ZA) i.e. impregnation, TVT treatment and direct injection. The structural, microstructural, and chemical bonding changes and densities of the final products were investigated and correlated to the experimental parameters. The authors provide sufficient literature review on the topic in the introduction part. The materials and methodology performed are almost sufficient. The work overall possesses partial novelty and results of this work are interesting in the field. The following issues should be addressed to improve the quality and integrity of the manuscript before its consideration for publication. Despite the raised the editorial changes that is appended to these comments with the hope of helping make the manuscript more integrated before its publication, the English of the manuscript needs further polishing. The discussion of the results looks scattered and confusing.
Here are my blind comments to the authors.
- It is not too clear how the current work is considerably different from the authors previous work since the reported additives, their mole ratios and obtained densities etc. are the same for some ZA additive cases? If same batches have been used and the research is incremented, this should be clearly stated. Crystals 2021, 11, 71. https://doi.org/10.3390/cryst11010071
- On line 31, the authors cite their previous 2 studies when referring to the modified SPS, however, none of their previous work have mentioned about any modification done to the commercial SPS and I do not see how these works are relevant to be cited here.
- Line 394, It is confusing how the powder sizes are increased if the solid-state mobility or mass diffusion is decreased.
- Line 394, what do the authors mean by improved crystal structure? The particle size cannot be related to the improved crystalline structure. There is no evidence of XRD showing increased crystalline size or improved degree of crystallinity or lower stacking faults etc. other than the authors have vague assumption of increased intensity of some peaks which can be easily related to preferred orientation especially considering the PS of the materials.
- Any insights on the 4-5 stages of densification and increased relative densities during the process as shown in Figure 5 would be useful.
- While the absorbance bands are discussed in the manuscript, the IR figure is shown in transmission mode, which might affect the interpretation of the results since conversion from absorption to transmission affects the intensities of the peaks not proportionally as per The Beer Lambert law. Has any baseline correction been performed on the spectra?
- The densities of the final compacts need to be double checked and compared with the reported values, most likely with Archimedes’ principle due to the measurement errors associated with the usage of calipers.
- The XRD patterns and the reflections should be indexed properly, i.e. the planes of the corresponding reflections etc. Also, what database has been used to confirm the phases?
- The particle sizes of the initial materials should be stated in the materials section. What is the supplier of the materials and how have the purity of the materials been tested?
- The phases of the reactants or products i.e. solid, liquid or gaseous states, should be indicated throughout the manuscript.
- The sentences listed below are not grammatically correct and confusing, thus need to be rephrased.
Lines 235-238, Line 98, Line 93.
- Line 328-329, Either chemical bond structure of all carboxylate, hydroxyls and others should be schematically shown or carboxylate should be removed since it has no help improving the manuscript nor to the results of discussion.
Author Response
Dear Reviewer, thank you for reading our manuscript and reviewing it, which will help us improve it to a better scientific level. In the commentaries, was noted “The discussion of the results looks scattered and confusing”. We revised our manuscript and the Discussion section has been edited. We hope that it will meet your expectations.
Reviewer 2 comment:
It is not too clear how the current work is considerably different from the authors previous work since the reported additives, their mole ratios and obtained densities etc. are the same for some ZA additive cases? If same batches have been used and the research is incremented, this should be clearly stated. Crystals 2021, 11, 71. https://doi.org/10.3390/cryst11010071
Authors´ answer
Line 103-110, 160-172
In previous work, studies were carried out on a simple setup consisting of a hydraulic press and a steel mold with induction heating. The sintering mode included prolonged heating and a long isothermal holding time, during which the sintering conditions could change significantly due to the extrusion of solutions from the working volume of the mold. The same ZnO reagent and other auxiliary reagents were used in the previous and new studies. Impregnated and autoclaved samples were prepared each time immediately before the studies.
Reviewer 2 comment:
On line 31, the authors cite their previous 2 studies when referring to the modified SPS, however, none of their previous work have mentioned about any modification done to the commercial SPS and I do not see how these works are relevant to be cited here.
Authors´ answer
It was corrected.
Reviewer 2 comment:
Line 394, It is confusing how the powder sizes are increased if the solid-state mobility or mass diffusion is decreased.
Authors´ answer
During a long period of TVT (20 hours in this work), the crystal structure of ZnO powder is transformed under the action of solid-phase mobility: the crystal size of powder samples increases. In addition, solid-phase mobility leads to a decrease in the number of defects in the crystal structure. As a result of the improvement of the crystal structure, its solid-phase mobility decreases, and crystal growth stops. The conclusion about improving the structure at a certain stage of TVT is made based on several signs. When crystals are formed under these conditions, the sizes of coherent scattering regions increase [39,40,41] the content of hydroxyl groups decreases [41,42,43,44] and well-defined crystals with flat sides and sharp edges appear [41,42]. Line 460-468.
Reviewer 2 comment:
Line 394, what do the authors mean by improved crystal structure? The particle size cannot be related to the improved crystalline structure. There is no evidence of XRD showing increased crystalline size or improved degree of crystallinity or lower stacking faults etc. other than the authors have vague assumption of increased intensity of some peaks which can be easily related to preferred orientation especially considering the PS of the materials.
Authors´ answer
At TVT in the presence of an activating additive, mobility of the structure appears due to exchange processes between the crystals and the medium. Mass transfer and crystal growth begins. In addition, the structure mobility leads to its refinement due to the reduction of defects. As a result of the perfection of the structure, its solid phase mobility decreases. Crystal growth stops. Conclusion about perfection of structure at a certain stage of TPO is made on the basis of several features. When crystals are formed under these conditions, the size of coherent scattering regions increases, the content of hydroxyl groups decreases, and well-defined crystal facets with smooth faces and sharp edges appear. These representations are added in the introduction. Line 97-110.
Reviewer 2 comment:
Any insights on the 4-5 stages of densification and increased relative densities during the process as shown in Figure 5 would be useful.
Authors´ answer
Our interpretation is presented in the Discussion section in the lines 405-407.
Reviewer 2 comment:
While the absorbance bands are discussed in the manuscript, the IR figure is shown in transmission mode, which might affect the interpretation of the results since conversion from absorption to transmission affects the intensities of the peaks not proportionally as per The Beer Lambert law. Has any baseline correction been performed on the spectra?
Authors´ answer
FT-IR spectra were taken in the standard diffuse reflectance mode, no baseline correction was required or performed. In Fig. 7 shows the reflection in relative units on the ordinate.
Reviewer 2 comment:
The densities of the final compacts need to be double checked and compared with the reported values, most likely with Archimedes’ principle due to the measurement errors associated with the usage of calipers.
Authors´ answer
The method proposed in [25] was chosen to control the density changes of the samples during CH. This method also allows us to determine the density of samples, for which measurements by the Archimedes method are impossible because of partial destruction and loss of their mass in the measuring environment.
Reviewer 2 comment:
The XRD patterns and the reflections should be indexed properly, i.e. the planes of the corresponding reflections etc. Also, what database has been used to confirm the phases?
Authors´ answer
Line 238-240.
The presence of layered basic zinc acetate (LBZA) is inferred from Tarat A., Nettle C.J., Bryant D.T.J., Jones D.R., Penny M.W., Brown R.A., Majitha R., Meissner K.E., Maffeis T.G.G. Microwave synthesis of layered nanosheets of basic zinc acetate and their thermal decomposition into nanocrystalline ZnO, Nanoscale Res Lett. 2014, V.9, pp.1-8.
Reviewer 2 comment:
The particle sizes of the initial materials should be stated in the materials section. What is the supplier of the materials and how have the purity of the materials been tested?
Authors´ answer
The crystal size of the initial ZnO powder and the supplier are added to the materials section (line 176). When investigating the effect of TVT and CSP on the morphology of ZnO crystals, it is important that when treated in aqueous medium without an activator, impurities in the powder do not cause changes. The zinc oxide used satisfied these requirements.
Reviewer 2 comment:
The phases of the reactants or products i.e. solid, liquid or gaseous states, should be indicated throughout the manuscript.
Authors´ answer
It was corrected.
Reviewer 2 comment:
The sentences listed below are not grammatically correct and confusing, thus need to be rephrased. Lines 235-238, Line 98, Line 93.
Authors´ answer
It was corrected.
Reviewer 2 comment:
Line 328-329, Either chemical bond structure of all carboxylate, hydroxyls and others should be schematically shown or carboxylate should be removed since it has no help improving the manuscript nor to the results of discussion.
Authors´ answer
Carboxylate structure removed.
Reviewer 3 Report
The manuscript by Ivakin et al. provides a detailed study of some of the approaches that can be used to make the cold sintering process (CSP) more amenable to industrial adaptation. Zinc oxide being one of the most widely studied materials for flash sintering, the paper builds on the existing data to help better understand and exploit the mechanism of CSP by using a combination of CSP and SPS (spark plasma sintering). By changing the microstructure and the nature of the surface of zinc oxide particles using zinc acetate dihydrate as an activator, a high density is achieved while reducing the amount of pressure and time required for the process. The microscopy and time dependent density measurements combined with structural information from XRD and DRIFTS, significantly further the mechanistic understanding of the processes and how it depends on the amount of activator and its introduction method, especially at shorter time scales of around five minutes. The paper establishes that the impregnation route leads to the highest density under the conditions of the CSP/SPS experiment due to creation of structural defects in the zinc oxide crystals during the impregnation step. The importance of the work is highlighted by the fact that it makes significant contributions to two aspects of the research in the field, improving the understanding of CSP and adapting it for practical applications.
While I find a lot of scientific merit in the work, the paper does have numerous issues that I feel need to be addressed before publication. With the high number of issues combined together, a major part of the paper needs to be revised. Below are my comments on these points.
Major comments:
- Extensive editing of English language and writing style is required throughout the text to make it more concise and fluid. The grammatical, typographical and labelling errors combined with awkward phrasing makes the paper difficult to follow. I have mentioned a few of these in the minor comments below and also included an annotated PDF copy of the manuscript with some more. However, the list is not exhaustive.
- In the introduction, the statement in lines 32-34 about the decrease in sintering time and temperature for cold sintering is a bit misleading. Both these effects are also major features of spark plasma and flash sintering. I suggest to modify this with appropriate references (eg. Ibn-Mohammed, T., C. A. Randall, K. B. Mustapha, J. Guo, J. Walker, S. Berbano, S. C. L. Koh, D. Wang, D. C. Sinclair, and I. M. Reaney. "Decarbonising ceramic manufacturing: A techno-economic analysis of energy efficient sintering technologies in the functional materials sector." Journal of the European Ceramic Society39, no. 16 (2019): 5213-5235).
- In line 89, the concept of activating additive is used without prior mention of it in the introduction. I suggest including a 1 or 2-line summary of what an activator is, its role in CSP and how it generally works, with appropriate references.
- It is not clear what the authors are trying to say in lines 98-99. It becomes clear only after reading the cited work [21]. However, without reading the cited work, it appears that the paper is comparing CSP vs. TVT as independent processes. It is only in lines 102-103 that it becomes clear that the comparison is between samples that underwent CSP either with or without a prior TVT step. Please clarify.
- It would help to include an SEM image and particle size distribution curve of the TVT_5% AC/ZnO sample in Figure 3.
- In Figure 3(d), it seems that the bimodal nature of the PSD curve as shown in Figure 3(d) and mentioned in lines 239-240 might be of significance as it can have a strong effect on packing and final density. This effect seems to be even more pronounced in the TVT_5%Ac/ZnO sample shown in Figure 4(b). Use of a single mean particle size in such a case obscures the finer effects of the particle size distribution. The authors are suggested to include at least a comment on the effect of this bimodal distribution on the density and how it compares to the crystallographic effects from impregnation or TVT during CSP/SPS.
- What are the general error ranges for the mean particle sizes reported in Table 1 using both the DLS method and SEM image analysis? The slight decrease in particle size mentioned in line 241 might not be statistically significant. Also this decrease in conflict with the statement in line 239 which says that the size range did not change.
- It will be helpful to include the PSD curves for the samples shown in Figure 4(a) and (b). Might help clarify the comments in point 6 mentioned above.
- To make all the data more digestible, I suggest including a short summary of the experimental plan at the start of the results section. For example, it would be helpful to talk about how the paper first looks at powder properties and the effects of various impregnation methods and activator concentrations on the powder, and the second set of data is for the cold sintered samples made from the powders mentioned in the first section. Without such a guide in the current manuscript, it becomes necessary to flip back and forth between pages to confirm whether the authors are talking about powders or sintered samples.
- The CS_ZnO+H2O-0.5%Ac sample in Figure 5 is introduced without any description and background in the text. The nature of the sample become clear only in Table 2 which is after the figure in the text.
- Line 293 mentions CS_0.5% Ac/ZnO, however, Figure 7 (a) shows a CS_ZnO+H2O sample. At the same time, line 294 mentions Figure 7(d) when there is no such figure in the paper. Please correct with appropriate sample data and figure numbers.
- In lines 388-389 the decrease in grain size is attributed to mass transfer inhibition by a basic acetate impurity phase. Are there any examples of this in prior work? If so, please cite.
- In Table 3, exactly which rows are for which samples from the previous tables and figure? Please specify with sample names. It also seems that rows 1 and 3 have the same conditions but different results. This needs to be corrected. When tracing back the samples to Table 2, the powder particle sizes in table 3 do not match.
Minor comments:
- Providing a comparison of the density of the non-impregnated sample with that of the impregnated sample in the abstract will provide a better idea of what to expect in the paper.
- In the introduction lines 103 and 104, please give examples of the traditional methods of activator addition from references.
- In lines 148-150, please include information about the manufacturers, grades etc. for the reagents used in the work.
- Please include manufacturer and model information for autoclave in line 162.
- The actual quantity of zinc oxide used needed in line 166 instead of ‘a quantity’.
- Pressure for pre-pressing needed in line 174 along with the manufacturer and model information for the Carver press.
- What was the thickness range for the pellets pressed in line 183?
- What was the pressure reached on vacuuming in the first step of SPS in line 188?
- In line 192, how does the pressure of 80 MPa compare with the threshold set up by Kang et al. (Kang, Xiaoyu, Richard Floyd, Sarah Lowum, Matthew Cabral, Elizabeth Dickey, and Jon‐Paul Maria. "Mechanism studies of hydrothermal cold sintering of zinc oxide at near room temperature." Journal of the American Ceramic Society102, no. 8 (2019): 4459-4469.)
- In lines 214-215, mention the measurement mode viz. secondary or backscattered.
- What was the time for ultrasonic deagglomeration in line 217?
- Please include manufacturer information for Image-Pro software in line 221.
- In the figure title in line 289, it needs to be clarified that the PSD is calculated from the respective SEM images.
- Need legend in graph in Figure 4(c).
- Please provide references for the traditional approaches to CSP mentioned in line 255.
- Line 274 mentions average size D, however Table 2 uses Mean grain size G. Please use uniform nomenclature.
- In lines 287-288, negligible effect of the sample CS_ZnO+H2O-0.5%Ac is with respect to the grain size, since there is actually a marked effect in terms of density when compared with the non-activated sample as shown in Table 2.
- It is not clear if the data for 30-60o in Figure 8 is the same for all experimental conditions. Please clarify.
- 5% Zn (Ac)2 mentioned in line 315 is not defined in the text prior. Please include a description or summarize all abbreviations in the beginning.
- The paragraph starting at line 314 seems to use ‘absorption’ and ‘adsorption’ interchangeably. Please correct.
- Clarify that the bands mentioned in line 318 are in addition to those observed for the initial ZnO powder.
- Line 321 mentions the width of the range for hydroxyl group vibrations as 3000 cm-1 what is the actual range that the next sentence refers to?
- In Figure 9, the legend says ‘TVT’ Is this TVT_5%Ac/ZnO? Need consistent nomenclature throughout the text.
- In line 342 I assume the authors meant to say ‘thermal treatment’ and not ‘thermocouple treatment’.
- In line 346 where it says ‘bands in the regions belonging to hydroxyl and carboxylate groups’ please also include the actual ranges.
- In line 387-388, it is not clear which samples and data from Table 2 the authors are talking about. Please specify the sample/s and also what the reference sample is.
- In line 394, where it says that the ‘crystal structure has improved’, the statement needs to be qualified based on the results and referring back to the appropriate figures.

Author Response
Dear reviewer, thank you for your constructive comments concerning our manuscript. We have studied your comments carefully. We have modified the manuscript which we hope meet with your approval.
Reviewer 3 comment:
Extensive editing of English language and writing style is required throughout the text to make it more concise and fluid. The grammatical, typographical and labelling errors combined with awkward phrasing makes the paper difficult to follow. I have mentioned a few of these in the minor comments below and also included an annotated PDF copy of the manuscript with some more. However, the list is not exhaustive.
Authors´ answer
The English language and writing style were carefully edited.
Reviewer 3 comment:
In the introduction, the statement in lines 32-34 about the decrease in sintering time and temperature for cold sintering is a bit misleading. Both these effects are also major features of spark plasma and flash sintering. I suggest to modify this with appropriate references (eg. Ibn-Mohammed, T., C. A. Randall, K. B. Mustapha, J. Guo, J. Walker, S. Berbano, S. C. L. Koh, D. Wang, D. C. Sinclair, and I. M. Reaney. "Decarbonising ceramic manufacturing: A techno-economic analysis of energy efficient sintering technologies in the functional materials sector." Journal of the European Ceramic Society39, no. 16 (2019): 5213-5235).
Authors´ answer
The wording was corrected.
Reviewer 3 comment:
In line 89, the concept of activating additive is used without prior mention of it in the introduction. I suggest including a 1 or 2-line summary of what an activator is, its role in CSP and how it generally works, with appropriate references.
Authors´ answer
Line 103-118. A description of the processes at TVT in the presence of the activator is added in the introduction.
In the literature devoted to the study of CSP, the additive in the reaction medium is usually considered as a component of the transport phase, through which mass transfer occurs by the dissolution-deposition mechanism [e.g., 25, 27].
Reviewer 3 comment:
It is not clear what the authors are trying to say in lines 98-99. It becomes clear only after reading the cited work [21]. However, without reading the cited work, it appears that the paper is comparing CSP vs. TVT as independent processes. It is only in lines 102-103 that it becomes clear that the comparison is between samples that underwent CSP either with or without a prior TVT step. Please clarify.
Authors´ answer
An explanation of our approach to the treatment of common process features in CSP and TVT is added in the introduction.
Reviewer 3 comment:
It would help to include an SEM image and particle size distribution curve of the TVT_5% AC/ZnO sample in Figure 3.
Authors´ answer
It was included. Now Figure 2.
Reviewer 3 comment:
In Figure 3(d), it seems that the bimodal nature of the PSD curve as shown in Figure 3(d) and mentioned in lines 239-240 might be of significance as it can have a strong effect on packing and final density. This effect seems to be even more pronounced in the TVT_5%Ac/ZnO sample shown in Figure 4(b). Use of a single mean particle size in such a case obscures the finer effects of the particle size distribution. The authors are suggested to include at least a comment on the effect of this bimodal distribution on the density and how it compares to the crystallographic effects from impregnation or TVT during CSP/SPS.
Authors´ answer
The appearance of two or more distribution components with different crystal sizes has little effect on the final density because there is no movement of the crystals for optimal spatial filling of their packing gaps. The final density depends on the redistribution of mass between crystals with gap filling due to the solid-phase mobility of the structure.
Reviewer 3 comment:
What are the general error ranges for the mean particle sizes reported in Table 1 using both the DLS method and SEM image analysis? The slight decrease in particle size mentioned in line 241 might not be statistically significant. Also this decrease in conflict with the statement in line 239 which says that the size range did not change.
Authors´ answer
The size range is not directly related to the average crystal size. The predominance of the distribution component with small crystals within the range leads to a decrease in the overall average crystal size.
Reviewer 3 comment:
It will be helpful to include the PSD curves for the samples shown in Figure 4(a) and (b). Might help clarify the comments in point 6 mentioned above.
Authors´ answer
The PSD curves was included.
Reviewer 3 comment:
To make all the data more digestible, I suggest including a short summary of the experimental plan at the start of the results section. For example, it would be helpful to talk about how the paper first looks at powder properties and the effects of various impregnation methods and activator concentrations on the powder, and the second set of data is for the cold sintered samples made from the powders mentioned in the first section. Without such a guide in the current manuscript, it becomes necessary to flip back and forth between pages to confirm whether the authors are talking about powders or sintered samples.
Authors´ answer
The plan was included. Line 253-259.
Reviewer 3 comment:
The CS_ZnO+H2O-0.5%Ac sample in Figure 5 is introduced without any description and background in the text. The nature of the sample become clear only in Table 2 which is after the figure in the text.
Authors´ answer
Corrected: the structure of the Results section has been changed and Figure 9 with the microstructure of the sample CS_ZnO + H2O-0.5% Ac is shown after Table 2.
Reviewer 3 comment:
Line 293 mentions CS_0.5% Ac/ZnO, however, Figure 7 (a) shows a CS_ZnO+H2O sample. At the same time, line 294 mentions Figure 7(d) when there is no such figure in the paper. Please correct with appropriate sample data and figure numbers.
Authors´ answer
It was corrected.
Reviewer 3 comment:
In lines 388-389 the decrease in grain size is attributed to mass transfer inhibition by a basic acetate impurity phase. Are there any examples of this in prior work? If so, please cite.
Authors´ answer
An example of the effect of the impurity phase on mass transfer is described in [Ivakin, Y., Smirnov, A., Kormilitsin, M., Kholodkova, A., Vasin, A., Kornyushin, M., Tarasovskii, V., Rybalchenko, V. Effect of mechanical pressure on zinc oxide recrystallization in aqueous medium during cold sintering. Russian J Phys. Chem. B, 2022. Article in press], which is so far available in Russian version.
Reviewer 3 comment:
In Table 3, exactly which rows are for which samples from the previous tables and figure? Please specify with sample names. It also seems that rows 1 and 3 have the same conditions but different results. This needs to be corrected. When tracing back the samples to Table 2, the powder particle sizes in table 3 do not match.
Authors´ answer
It was corrected.
Reviewer 3 comment:
Providing a comparison of the density of the non-impregnated sample with that of the impregnated sample in the abstract will provide a better idea of what to expect in the paper.
Authors´ answer
It was corrected.
Reviewer 3 comment:
In the introduction lines 103 and 104, please give examples of the traditional methods of activator addition from references.
Authors´ answer
references added.
Reviewer 3 comment:
In lines 148-150, please include information about the manufacturers, grades etc. for the reagents used in the work.
Authors´ answer
The information about the manufacturers, grades was included.
Reviewer 3 comment:
Please include manufacturer and model information for autoclave in line 162.
Authors´ answer
The information was included. Line 189.
Reviewer 3 comment:
The actual quantity of zinc oxide used needed in line 166 instead of ‘a quantity’.
Authors´ answer
It was corrected.
Reviewer 3 comment:
Pressure for pre-pressing needed in line 174 along with the manufacturer and model information for the Carver press.
Authors´ answer
The information was added.
Reviewer 3 comment:
What was the thickness range for the pellets pressed in line 183?
Authors´ answer
The range was 0.34 – 0.50 mm. Line 210.
Reviewer 3 comment:
What was the pressure reached on vacuuming in the first step of SPS in line 188?
Authors´ answer
The information was added.
Reviewer 3 comment:
What was the pressure reached on vacuuming in the first step of SPS in line 188?
Authors´ answer
The information was added. Line 214.
Reviewer 3 comment:
In line 192, how does the pressure of 80 MPa compare with the threshold set up by Kang et al. (Kang, Xiaoyu, Richard Floyd, Sarah Lowum, Matthew Cabral, Elizabeth Dickey, and Jon‐Paul Maria. "Mechanism studies of hydrothermal cold sintering of zinc oxide at near room temperature." Journal of the American Ceramic Society102, no. 8 (2019): 4459-4469.)
Authors´ answer
The pressure of 80 MPa at 250°C is much lower than the threshold of transition to hydrothermal conditions mentioned in the mentioned article (it can be seen in Fig. 9 of Kang et al.). Moreover, the design of the graphite mold with graphite foil is leaky and most of the water is squeezed out of the working volume so that the sintering takes place mainly in the environment of water vapor, i.e. in thermovapour conditions.
Reviewer 3 comment:
In lines 214-215, mention the measurement mode viz. secondary or backscattered.
Authors´ answer
The information was added. Line 243-244.
Reviewer 3 comment:
What was the time for ultrasonic deagglomeration in line 217?
Authors´ answer
The information was added. Line 246.
Reviewer 3 comment:
Please include manufacturer information for Image-Pro software in line 221.
Authors´ answer
The information was added. Image-Pro software (Media Cybernetics, USA)
Reviewer 3 comment:
In the figure title in line 289, it needs to be clarified that the PSD is calculated from the respective SEM images.
Authors´ answer
Clarifications have been added to the captions of all drawings that show PSDs.
Reviewer 3 comment:
Need legend in graph in Figure 4(c).
Authors´ answer
The legend was added.
Reviewer 3 comment:
Please provide references for the traditional approaches to CSP mentioned in line 255.
Authors´ answer
The references were added.
Reviewer 3 comment:
Line 274 mentions average size D, however Table 2 uses Mean grain size G. Please use uniform nomenclature.
Authors´ answer
It was corrected.
Reviewer 3 comment:
In lines 287-288, negligible effect of the sample CS_ZnO+H2O-0.5%Ac is with respect to the grain size, since there is actually a marked effect in terms of density when compared with the non-activated sample as shown in Table 2.
Authors´ answer
Thank you for the comment. It is taken into account and added to the text. Line 380-381.
Reviewer 3 comment:
It is not clear if the data for 30-60o in Figure 8 is the same for all experimental conditions. Please clarify.
Authors´ answer
The data are the same. A clarification has been added to the text. Line 316-317.
Reviewer 3 comment:
5% Zn (Ac)2 mentioned in line 315 is not defined in the text prior. Please include a description or summarize all abbreviations in the beginning.
Authors´ answer
It was corrected. Line 323.
Reviewer 3 comment:
The paragraph starting at line 314 seems to use ‘absorption’ and ‘adsorption’ interchangeably. Please correct.
Authors´ answer
The term absorption is used when characterizing the band of DRIFTS spectrum. The term adsorption refers to the interaction of molecules with the ZnO surface. Both terms are used according to their definitions.
Reviewer 3 comment:
Clarify that the bands mentioned in line 318 are in addition to those observed for the initial ZnO powder.
Authors´ answer
Андрей, при переводе выпала фраза
The application of zinc acetate by the impregnation method leads to the appearance of several absorption bands in the DRIFTS spectra of ZnO samples, which were absent from the original ZnO powder.
Reviewer 3 comment:
Line 321 mentions the width of the range for hydroxyl group vibrations as 3000 cm-1 what is the actual range that the next sentence refers to?
Authors´ answer
Valence vibrations of hydroxyl groups cause the absorption in a wide area with a maximum of about 3363 cm-1.
Reviewer 3 comment:
In Figure 9, the legend says ‘TVT’ Is this TVT_5%Ac/ZnO? Need consistent nomenclature throughout the text.
Authors´ answer
It was corrected.
Reviewer 3 comment:
In line 342 I assume the authors meant to say ‘thermal treatment’ and not ‘thermocouple treatment’.
Authors´ answer
It was corrected to thermovapour treatment.
Reviewer 3 comment:
In line 346 where it says ‘bands in the regions belonging to hydroxyl and carboxylate groups’ please also include the actual ranges.
Authors´ answer
Included in line 357.
Reviewer 3 comment:
In line 387-388, it is not clear which samples and data from Table 2 the authors are talking about. Please specify the sample/s and also what the reference sample is.
Authors´ answer
Samples is specified in line 454.
Reviewer 3 comment:
In line 394, where it says that the ‘crystal structure has improved’, the statement needs to be qualified based on the results and referring back to the appropriate figures.
Authors´ answer
It was corrected. Line 460-468.
Round 2
Reviewer 2 Report
The authors have addressed my questions and comments in a satisfactory manner. I recommend the publication of this study. Thanks to all authors for their contribution to science.
Reviewer 3 Report
The authors have addressed all of the comments satisfactorily. Given the significance of the content and interest to the wider field, I recommend accepting the paper in its present form pending any minor spell check corrections.